# Can the Rational Design of International Institutions Solve Cooperation Problems? Insights from a Systematic Literature Review

Adela Toscano-Valle [1], Antonio Sianes [1], Francisco Santos-Carrillo [2] and Luis A. Fernández-Portillo [3,*]

1 Research Institute on Policies for Social Transformation, Universidad Loyola Andalucía, 14004 Cordoba, Spain; atoscanovalle@al.uloyola.es (A.T.-V.); asianes@uloyola.es (A.S.)
2 Department of International Studies, Universidad Loyola Andalucía, 14004 Cordoba, Spain; frsantos@uloyola.es
3 Department of Business Management, Universidad Loyola Andalucía, 14004 Cordoba, Spain
* Correspondence: portillo@uloyola.es

**Abstract:** Global governance challenges highlight the role of international institutions as problem-solving structures. Institutional design is, more than ever, relevant in this context. The academic literature on this issue is characterized by the existence of consolidated debates such as that of rationalism vs. constructivism, with a focus on making specific contributions to the rational design of international institutions. Koremenos, Lipson, and Snidal (2001) elaborated upon a series of cause-effect conjectures linking cooperation problems, considered independent variables, with institutional design features, considered dependent variables. This research aims to highlight the empirical evidence of the existing debate on this work by conducting a systematic review. Twenty-one quantitative research studies were collected through a screening and selection procedure and were subject to systematization. The findings showed asymmetric approaches to the rational design project, and agreements were the type of international institution that received the most attention from academia. Rationalism was supported by most of the body of literature. However, a broad subgroup of articles complemented this rational approach with other variables or schools of thought, such as those of constructivism and historical institutionalism. The results have relevance for the international institution design literature, as future avenues of potential research are underlined.

**Keywords:** cooperation problems; rational design; international institutions; global governance; constructivism; systematic review

## 1. Introduction

One of the most relevant and perceptible transformations in societies in recent decades has been the proliferation of transnational policies, whose formulation and administration take place in governance spaces that go beyond the frameworks of national sovereignty. The definition of these policies involves a conglomerate of international institutions and non-governmental actors that accompany states and contribute to solving the problems of growing global interdependence [1]. However, in contrast to the certainties of authority, legitimacy, legality, and orchestration capacity offered by national governance arenas, transnational agreements and commitments have ill-defined boundaries and uncertain guarantees of compliance. They belong to the realm of global governance.

According to Koremenos, Lipson, and Snidal [2], international institutions are "explicit arrangements, negotiated among international actors, that prescribe, proscribe, and/or authorize behaviour". They are rational responses to problems of cooperation, conscious creations that aim to solve collective action and cooperation problems through negotiation and contract between the actors involved, even if imperfectly [2–4]. This perspective provides a possible answer to one of the most prominent puzzles in the study of international

institutions: the variety and diversity of designs they register and their relationship with aspects such as scope, outcomes, and effects.

Understood as a set of rules, international institutions are designed to govern international behaviour [5]. These rules are produced through three governance mechanisms. First, through international organizations, which states endow with legal authority and legitimacy. Second, non-governmental actors, on their own or in alliance with international organizations and other governmental actors, also establish international agreements in the form of regulatory regimes and frameworks for the materialisation of transnational enterprises and objectives. Finally, states voluntarily disseminate international rules for the implementation of domestic policies. The latter case, while design challenges remain a relevant factor in adapting or adjusting to domestic patterns, poses fewer problems in terms of authority [6]. This paper will focus on the first two types. In this respect, international organizations are a more legitimized and legalized—i.e., more institutionalized—subset than a mere international agreement. The design of variables such as control, centralization, or flexibility can be decisive for policy outcomes.

International agreements and organizations shape the processes and spaces where transnational policies take place through the production of formal and informal norms and rules. These processes can sometimes challenge established political authority, influencing the policy process itself by altering the behaviour of actors whose loyalty and bargaining strategy must adapt to the new scenario. Institutional designs thus become a key factor in determining policy effectiveness and efficiency. For example, a poor institutional design of the 2030 Agenda (an agreement) has influenced the effectiveness of the international reaction to COVID-19 because, for instance, according to this design, it was rational for some states to defect from cooperation [7].

Within this framework, design possibilities are manifold, depending on the strategically prioritised choices of the actors involved: from national actors shaping preferences and pushing governments towards one or another design; transnational actors aiming to overcome the constraints of national regulations; and states prioritising their geopolitical interests and relative gains rather than responding effectively or efficiently to cooperation problems.

The model of Koremenos, Lipson, and Snidal [2] is one of the most prominent in the literature. It explains design variation in aspects such as membership, scope, centralization, control, or flexibility not only as a functional and strategic response to problems of distribution and implementation, but also to problems related to the number and nature of participating actors and to uncertainty about actors' behaviour and preferences and about the state of the world.

Since their contribution is embedded in the rational choice institutionalism [3,4,8–11], criticism comes from those who understand the behaviour of institutions as historically defined and socially constructed dynamic processes, along which transformations related to the identity, behaviour, or interests of the actors may occur [12]. Hence, institutional design is approached from multiple perspectives, each of which stress certain driving principles of cooperation: power, efficiency and rationality, moral values, shared history, common ideas, etc. These schools of thought differ in terms of their understanding of organizational behaviour and the processes by which institutions emerge and evolve.

Constructivist approaches claim that institutions are endogenous and collective bodies of shared norms, values and ideas that guide behaviour and shape identities. Hence, constructivist authors have long presented empirical evidence highlighting the importance of identity, bureaucracies, the diffusion of institutional frameworks by emulation, and other moral and adequacy considerations on the collective action of international actors [13–26]. Other questions refer to the omission of ideational aspects, not strictly functional or utility aspects, such as those related to the legitimacy of international norms [27,28]. In a similar vein, historical approach [29–31] emphasizes the role of contingent factors as the conscience collective in the configuration of institutions.

Today, the greatest theoretical dichotomy involves rationalism against constructivism [32], namely, the logic of calculation against the logic of appropriateness [27,28]. This research

contributes to this debate by questioning the validity of a prominent rational choice contribution and by observing how and to what extent rational authors have explained institutional design. The results distil some insights but also emphasize this need for a cohesive dialogue between the adherents of customarily opposed traditions.

It is not the aim of this contribution to delve into the long-lasting debate on which is the "valid" approach to institutional design and even less to solve it. There is evidence in the literature that supports the idea that, in some cases, institutions are not rationally designed (notably Allee and Elsig [33]), but there is also strong support for the rational approach, as this paper tries to show.

One single perspective, perhaps, is not able to face the current international challenges [12]. It could be the case that some institutions are designed as a result of a combination of different approaches, in which rationalism usually plays an important role, as Reinsberg and Westerwinter [34] state. Indeed, Finnemore and Sikkink [21] (p. 888) would theorize about a "strategic social construction" behind political processes. Whereas constructivism offers a sound explanation of how international institutions are created, the rational approach might better explain their subsequent evolution and reform [27,35,36]. In this line, analytical frameworks recently proposed have bridged rational choice with constructivist [37] and historical [38] institutionalism.

Despite its flaws, rationalism provides a framework that is able to shed light on the optimal design of institutions, and, most remarkably, to propose empirically testable hypotheses about how institutions should be designed (or remodelled) to improve cooperation and better address social problems [12].

Nevertheless, it is evident that many institutions have not been designed to solve these problems, but they emerged as a result of political compromises. Even in this case, the rational design approach helps us understand how these institutions operate [39].

## 2. Theoretical Framework

International regime theories and (neo) liberal institutionalism succeed to offer a rational response to the proliferation of international institutions, a challenge to hegemonic stability theories [40–42]. The creation of international institutions was not solely a function of the power and interests of hegemonic powers, but a balancing and bargaining game based on cooperation on the basis of common rules, formal and informal, for mutual benefit [43–45]. Under this premise, institutional design becomes particularly relevant because it characterizes and defines the mechanisms that facilitate the enforcement and effectiveness of agreements and treaties [8]. Rational choice authors assume the inherent and universal rationality of actors operating in the international system. Designers of international institutions weigh cost and benefits and facilitate decision-making procedures that lead to optimal outcomes. Game theory, and mathematical and economic approaches are useful tools to explain this set of problems.

Early rationalist approaches analyzed compliance problems arising from cooperation [46–48]. They focused on changes in the conception of sovereignty [49] on the asymmetric benefits of principal–agent models [48,50] also on aspects of formalization and legalization of rules [3,51,52]. Koremenos, Lipson, and Snidal [2] focus on the analysis of collective action problems and incomplete information.

The specific contribution on the rational design project by Koremenos, Lipson, and Snidal [2] is one of the most prominent in recent literature [5,12,53]. According to Koremenos, Lipson, and Snidal [2], institutional diversity cannot be random since actors are rational and their interactions are objective-oriented. If states build and shape institutions to face cooperation problems, the organizational features of these institutions should be able to be explained by these challenges. Hence, these authors develop a causal analytical framework in which cooperation problems, which are treated as independent variables, impact institutional design features, which are treated as dependent variables. The relationships between these variables are used to develop sixteen different hypotheses.

Within this analytical framework, the institutional design features are characterized by five dependent variables. They are not the only dimensions of institutional design that exist, but they represent its prominent and easily measurable issues. Koremenos, Lipson, and Snidal [2] synthesized these features, as shown in Table 1.

**Table 1.** Definitions of the dependent variables.

| Institutional Dimensions | |
| --- | --- |
| Membership | Rules determining who belongs to the institution |
| Scope | Broad or narrow spectrum of issues covered and whether they are linked |
| Centralization | Formalization or performance of important tasks by a single focal entity |
| Control | Voting and decision-making rules that determine collective decisions |
| Flexibility | Accommodation procedures used to adapt to new circumstances |

Source: The authors, based on Koremenos, Lipson, and Snidal [2].

As international institutions are rational constructions that are the result of the determination of states to pursue both individual and collective goals, international actors must overcome inherent cooperation problems to ensure effective cooperation. Consequently, Koremenos, Lipson, and Snidal [2] define cooperation problems as the independent variable influencing the specific institutional design employed by such actors. Cooperation problems correspond to four independent realities, as summarized in Table 2. It is worth mentioning that later contributions like Koremenos [54] would add some variables to this initial model. However, we focus on the Koremenos, Lipson, and Snidal [2] framework because we aim, precisely, to observe its later contestation.

**Table 2.** Definitions of the independent variables.

| Cooperation Problems | |
| --- | --- |
| Distribution problems | Appear when actors have different preferred alternatives or outcomes |
| Enforcement problems | Appear when actors have individual incentives to defect. Actors sacrifice long-term cooperation for the current unilateral benefits of non-cooperation |
| Number | Number of actors potentially relevant to joint welfare and asymmetries among them |
| Uncertainty | Appears when there is imperfect or incomplete information regarding the three strategic elements of choices, consequences, and preferences. As such, three different types of uncertainty are described: uncertainty about behaviour, uncertainty about the state of the world and uncertainty about preferences |

Source: The authors, based on Koremenos, Lipson, and Snidal [2].

Once the dependent and independent variables have been identified, the cornerstone of this model can be developed: rational design conjectures. Four general assumptions or premises form the basis of these conjectures, which are used to explain and understand institutional design from a rationalist point of view; these premises relate to rational design, the shadow of the future, transaction costs and risk aversion. All things considered, a total of sixteen different conjectures have been identified by Koremenos, Lipson, and Snidal [2]. Table 3 summarizes these conjectures, most of which show positive correlations, where both variables vary in the same direction, while two of them, V1 and F3, exhibit the inverse.

Table 4 shows the interconnection of these variables and conjecture outcomes in a more intuitive way.

Criticism of this model has been wide-ranging and diverse since its dissemination. It points to limitations both in scope and in analytical and empirical frameworks. A first criticism refers to the limited conception of institutions it offers. There are other possible rational approaches, and they are even capable of incorporating contingent factors of social and historical contexts [55]. Failure to consider these alternative aspects detracts from its causal depth [27]. More recent work has extended the rational framework along these lines [26]. On the other hand, the limitations of the model are also perceived empirically, since not all institutions can be studied using this model [56]. Indeed, some typologies of institutions do not respond to the main premises of the model, for example, those of an ideological nature or those that are established under some kind of imposition.

**Table 3.** Summary of Rational Design conjectures.

| | |
|---|---|
| M1 | Restrictive Membership increases with the severity of the Enforcement problem |
| M2 | Restrictive Membership increases with the Uncertainty about preferences |
| M3 | Membership increases with the severity of the Distribution problem |
| S1 | Scope increases with Number |
| S2 | Scope increases with the severity of the Distribution problem |
| S3 | Scope increases with the severity of Enforcement problem |
| C1 | Centralization increases with Uncertainty about behaviour |
| C2 | Centralization increases with Uncertainty about the state of the world |
| C3 | Centralization increases with Number |
| C4 | Centralization increases with the severity of the Enforcement problem |
| V1 | Control decreases with Number |
| V2 | Asymmetry of Control increases with asymmetry of contributors (Number) |
| V3 | Control increases with Uncertainty about the state of the world |
| F1 | Flexibility increases with Uncertainty about the state of the world |
| F2 | Flexibility increases with the severity of the Distribution problem |
| F3 | Flexibility decreases with Number |

Source: Koremenos, Lipson, and Snidal [2] (p. 797) modified by authors.

**Table 4.** Conjectures Resulting from Variable Intersections.

| | Institutional Dimensions | | | | |
|---|---|---|---|---|---|
| **Cooperation Problems** | **Membership** | **Scope** | **Centralization** | **Control** | **Flexibility** |
| Distribution | M3 | S2 | | | F2 |
| Enforcement | M1 | S3 | C4 | | |
| Number | | S1 | C3 | V1 | F3 |
| | | | | V2 | |
| Uncertainty | | | | | |
| About behaviour | | | C1 | | |
| About the state of the world | | | C2 | V3 | F1 |
| About preferences | M2 | | | | |

Source: Koremenos, Lipson, and Snidal [2].

A second important critical argument highlights the limitations of studying how design influences the effectiveness of institutions. The model of Koremenos, Lipson, and Snidal [2] offers an explanation of the causes and variations of design, i.e., creation and maintenance. In short, they do not explain outcomes and effects on states' behaviour. Such divergence implies that research on institutional design can be divided into two different approaches. One that would focus on the causes, factors that explain how (dependent) institutions are configured. Other that would research on the consequences, diving in the significance of a certain (independent) institutional design.

Other criticisms point to the approach's omissions of the dynamic aspects of institutions, especially those related to task variation and power, the relationship with domestic politics or the control that bureaucracies exert over them.

At this point, it is important to note that the primary purpose of the work of Koremenos, Lipson, and Snidal is to "generate testable propositions that will guide the empirical analysis of international institutions" [2] (p. 782). Twenty years later, questions remain regarding whether and how this objective has been accomplished, as the discussion on this topic has mainly been driven by normative approaches. This study, however, relies on an empirical approach, systematizing the contributions and conclusions of the empirical studies that have examined this issue. The works in this body of literature widely differ in terms of their methodological approaches: some perform case studies or comparative research, and others develop genuine quantitative studies. This research focuses on the latter, as it embraces the empirical analysis performed by Koremenos, Lipson, and Snidal [2]. Indeed, Reinsberg and Westerwinter [57] (p. 60) would claim that "large-N analysis is the only way to arbitrate among different explanations of institutional design".

The proliferation of quantitative analyses that aim to combine theoretical and empirical issues has revitalized the research on the institutional design of international institutions. The emergence of large datasets was crucial for these advances [2,51]. As Westerwinter [57] (p. 140) points out, the emergence of large-n data sets related to international organizations and institutional arrangements has allowed for progress to be made towards establishing a body of work based on quantitative analysis that can revitalize the research in this field [3,54,58–64].

This research aims to observe how quantitative research studies based on large-n datasets and embedded within the rational design approach have contributed to the main debate on international institutions. It aims to reveal how their statements have been endogenously (by the authors' own perspectives) and externally (by other authors' views) contested. In this study, the methodological approach selected is a systematic literature review. This technique captures qualitative information with the objectivity of a systematic technique. Based on data obtained from the Web of Science, this research systematizes a set of 161 documents that are scanned and filtered based on approved selection criteria to identify the final population of 21 documents considered. The results of this systematization shed light on the abovementioned main theoretical debates, on the capacity of this methodology to contribute to the resolution of these debates, and on its ability to contribute to the resolution of the main problems related to international cooperation in times of crisis. The conclusions can provide insights for decision-makers, for the academic literature on global governance and regarding the design of international institutions.

## 3. Materials and Methods

As previously stated, a systematic review was selected as this study's methodology. This methodological approach is similar to a literature review but differs from a purely narrative approximation. This explains why systematic reviews are especially widespread in the medical field [65] and used to support evidence-based medicine (Sackett 1997). However, the objectivity and systemic procedure of this method have proven to be sound for research within the social science domain; thus, the use of this method has emerged in many fields [66], including those of international relations [67,68] and international development [69].

The widespread use of systematic reviews is explained by the inherent advantages of any systematic analysis: objectivity, validity and rigour. To successfully capitalize upon these advantages, it is crucial to carefully follow the steps of such a procedure. Thus, the Cochrane Handbook for Systematic Reviews of Interventions [70] was used and the Preferred Reporting Items for Systematic Reviews and Meta-Analyses Protocols (PRISMA-P) [71] (The PRISMA checklist is available as Supplementary Materials), being the protocol not explicitly described. The systematic review protocol is in the process of registration in the International Prospective Register of Systematic Reviews PROSPERO (http://www.crd.york.ac.uk/PROSPERO) from 9 April 2022.

The point at which we departed from this method was the identification of publications within the Web of Science database. Recent studies have relied on this repository because it contains the most prominent journals of every discipline [72,73]. To retrieve publications from the Web of Science, a search vector is needed. Ours consisted of three juxtaposed sets: a unit of analysis, an object of analysis, and the methodological approach of interest. The unit of analysis was "international institutions" in a broad sense; this included international agreements, international organizations, and governance initiatives. The object of analysis was the rational design of international institutions identified by Koremenos, Lipson, and Snidal [2]. Therefore, author references and "rational design" synonyms were included. Finally, key terms were introduced to ensure that the examined studies used the desired methodological approaches: empirical and quantitative analyses built upon datasets. Research vector can be found in the Appendix A.

On 8 April 2020, the search was conducted on the Web of Science platform, and 161 research studies (N = 161) written in English and published since the introduction of the

examined model were retrieved. After their retrieval, the studies underwent a screening procedure based on their titles and abstracts. Both the procedure and the final output were double-checked by two other independent researchers (L. F-P and F.S.-C.). A set of inclusion and exclusion criteria was defined in advance. The criteria were arranged consecutively to construct a strainer, namely, a tool that progressively refines an original search. The result is a list of the 21 research articles considered in the systematic review is shown in Table 5. A detailed graph of these procedures and the criteria can be found in the Appendix A.

**Table 5.** Articles considered in the systematic review in alphabetical order of authors' names.

| Code | Referecne | Title | Journal |
|------|-----------|-------|---------|
| 001 | Allee and Elsig [74] | Why do some international institutions contain strong dispute settlement provisions? New evidence from preferential trade agreements | Review of International Organizations |
| 002 | Baccini, Dür, and Elsig [64] | The Politics of Trade Agreement Design: Revisiting the Depth-Flexibility Nexus | International Studies Quarterly |
| 003 | Bearce, Eldredge, and Jolliff [75] | Do Finite Duration Provisions Reduce International Bargaining Delay? | International Organization |
| 004 | Bernauer et al. [76] | Is there a "Depth versus Participation" dilemma in international cooperation? | Review of International Organizations |
| 005 | Blake and Payton [77] | Balancing design objectives: Analyzing new data on voting rules in intergovernmental organizations | Review of International Organizations |
| 006 | Copelovitch and Putnam [78] | Design in Context: Existing International Agreements and New Cooperation | International Organization |
| 007 | Hansen, McLaughlin Mitchell, and Nemeth [79] | IO mediation of interstate conflicts—Moving beyond the global versus regional dichotomy | Journal of Conflict Resolution |
| 008 | Hooghe and Marks [63] | Delegation and pooling in international organizations | Review of International Organizations |
| 009 | Jetschke and Münch [80] | The Existence of Courts and Parliaments in Regional Organizations: A Case of Democratic Control? | Politische Vierteljahresschrift |
| 010 | Jo and Namgung [81] | Dispute Settlement Mechanisms in Preferential Trade Agreements: Democracy, Boilerplates, and the Multilateral Trade Regime | Journal of Conflict Resolution |
| 011 | Kaoutzanis, Poast, and Urpelainen [82] | Not letting 'bad apples' spoil the bunch: Democratization and strict international organization accession rules | Review of International Organizations |
| 012 | Koremenos [36] | Contracting around international uncertainty | American Political Science Review |
| 013 | Koremenos [83] | If only half of international agreements have dispute resolution provisions, which half needs explaining? | Journal of Legal Studies |
| 014 | Koremenos [84] | The Continent of International Law | Journal of Conflict Resolution |
| 015 | Koremenos [85] | What's left out and why? Informal provisions in formal international law | Review of International Organizations |
| 016 | Kucik [86] | The Domestic Politics of Institutional Design: Producer Preferences over Trade Agreement Rules | Economics & Politics |
| 017 | Lefler [87] | Strategic forum selection and compliance in interstate dispute resolution | Conflict Management and Peace Science |
| 018 | Marcoux [88] | Institutional Flexibility in the Design of Multilateral Environmental Agreements | Conflict Management and Peace Science |
| 019 | Mohrenberg, Koubi, and Bernauer [89] | Effects of funding mechanisms on participation in multilateral environmental agreements | International Environmental Agreements—Politics Law and Economics |
| 020 | Tallberg, Sommerer, and Squatrito [62] | Democratic memberships in international organizations: Sources of institutional design | Review of International Organizations |
| 021 | Tir and Stinnett [90] | The Institutional Design of Riparian Treaties: The Role of River Issues | Journal of Conflict Resolution |

Regarding the methodology for the qualitative analysis, the systematization follows the abovementioned triangular logic of reasoning involving the object, unit, and method of analysis. The previously established observation procedure ensures that qualitative

information is recollected objectively. Since this procedure is the cornerstone of the investigation, the full text of the examined records must be carefully analysed in a standardized manner. By evaluating the examined authors' statements methodically, this research is able to identify the patterns and gaps within the rational literature on institutional design. Observed items are found in Table 6 below.

**Table 6.** Axes of systematization within the systematic review.

| Unit of Analysis | Object of Analysis | Methodological Approach |
| --- | --- | --- |
| Subject of analysis | Relevant hypotheses | Time frame |
| Type of institution | Koremenos approach or application | Number of cases |
| Issue area | Dependent/independent variables | Research objectives |
| Organizational scope | Koremenos conjectures | Statistical method |
| | Implications for Rational Design | Dataset source |

## 4. Results and Discussion

### 4.1. Unit of Analysis

The unit of analysis step focused on the specific features of the international institutions analyzed in the body of literature. Thus, the first result of our review categorizes the records by the formalization degree of each analyzed international institution, and this categorization revealed an asymmetric distribution. Contrary to what may be expected, given the more formal approach of rational and functionalist authors towards international institutions, international agreements were the institutions most often analyzed by the examined academic literature. The systematization showed that 67% of the scholars performed empirical research on international agreements, while only 33% explored international organizations. Consequently, this research highlights the need for increased empirical attention regarding the design of international organizations.

Among international agreement research studies, one can observe an internal divergence stemming from the reach of such agreements. Some authors focus on sectoral agreements such as preferential trade agreements and environmental agreements, while others focus on those regulating sovereignty, such as River treaties; still others analyze multi-issue agreements. As shown in Figure 1, multi-issue international agreements are the most widely represented in the examined body of literature. Regarding specific agreements, the designs of Preferential trade agreements and Environmental agreements receive major attention. Thus, there are insufficiently explored paths of research regarding human rights and security agreements.

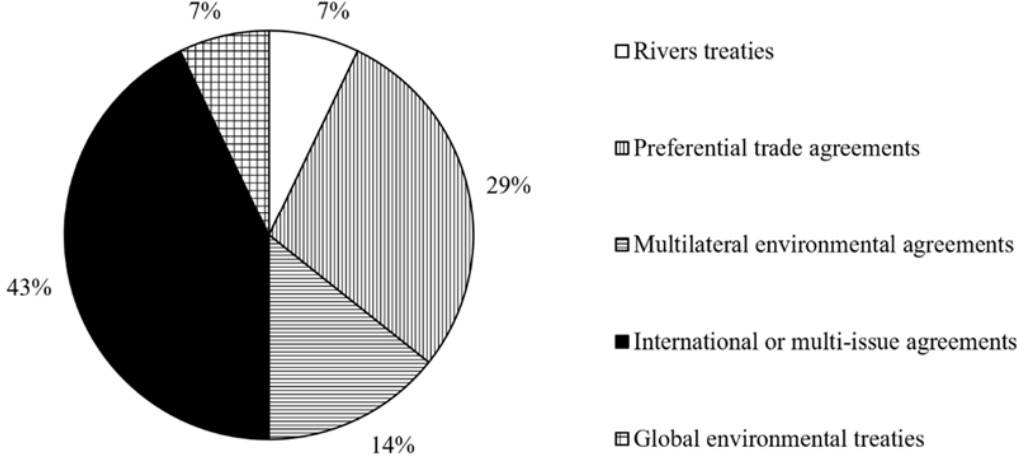

**Figure 1.** Types of agreements in the reviewed documents.

All the international agreement studies are either authored by Koremenos alone [36,83–85] or find support in her research papers published since 2005 [75,78]. This fact is consistent

with this scholar's powerful research program, the Continent of International Law (COIL). The COIL database is built on the belief that issue heterogeneity is intrinsic to the concept of international institution design. Of course, studies using a COIL dataset perform empirical research on multi-issue international agreements. The COIL research program goes a step further than Koremenos, Lipson, and Snidal's [2] original work. Indeed, it is a recent contribution by Koremenos that has encouraged and guided research. To this point, the systematization of this study suggests that rationalism has explanatory power in the context of the design of multi-issue international agreements.

On the other hand, within the field of international organizations, research studies differ based on organizational scope. As shown in Figure 2, authors have studied intergovernmental, international, and regional organizations. Remarkably, Hansen, McLaughlin Mitchell, and Nemeth [79] studied both international and regional organizations using the same dataset.

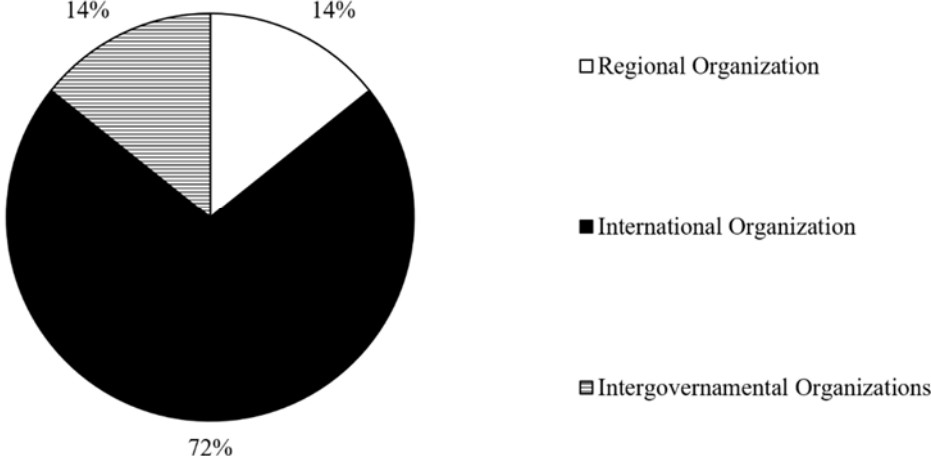

**Figure 2.** Types of organizations in the reviewed documents.

The lack of quantitative research studies arguing for the rational design of international organizations suggests two things. First, the rational school of thought needs to conduct further empirical research regarding international organizations. Second, complementary dialogue between different schools of thought could offer a more complete perspective on the design of various international institutions.

### 4.2. Methodological Approach

A systematization can reveal the synergy within the empirical research on the rational design of international institutions. In other words, primary sources of information are rare. Studies support secondary or, ultimately, mixed sources of information. The COIL project led by Koremenos and the Issue Correlates of War (ICOW) led by the Pevehouse research team were observed to be significantly recurrent. On the other hand, other researchers offer novel dataset contributions [64,80,86,89,90]. These cases illustrate the way in which the examined discipline was built. Future studies that want to contest rationalism and employ other perspectives such as constructivism may find this information particularly useful. If recurrent rationalist datasets were employed in alternative investigations, the outcomes would be especially productive.

Contrary to the expectation of divergence regarding time frame, all the contributions were categorized as cross-sectional studies. Even though some datasets covered a longitudinal time period, institutional evolution over time was not central to these investigations. The fact that all the research studies were cross-sectional may be explained by certain foundational rational approach principles. Longitudinal studies are suitable when time is a variable of interest and hypotheses are built upon that variable. As one might recognize, this concern is consistent with the historical institutionalism perspective but is useless from the rationalist point of view. The fact that time is never a variable of interest when

adopting the rational approach highlights the main critique of this method: its inability to incorporate factors as path dependence. Consequently, this systematization highlighted the need for a complementary dialogue between different traditions regarding the design of international institutions.

General patterns were also identified in the context of methodological techniques. When the objectives of the studies were examined, a unified outcome emerged: all the articles were identified as correlation studies. Because the Koremenos, Lipson, and Snidal [2] model specifically formulated conjectures as correlated propositions, this characteristic was certainly expected. However, rationalist principles may also explain why authors hypothesize about variable interrelations. Scholarship based on rational design treats institutions as exogenous constructions built by states. Thus, actors are thought to be influenced by external variables. This statement fundamentally differs from the propositions of constructivist authors, who view international institutions as endogenous creations. Therefore, further research identifying constructivists' research objectives would be especially enriching.

Regarding the mathematical methods employed for the testing of the examined hypotheses, different statistical techniques were found. Based on the hypotheses that they examined, the authors utilized bivariate and multivariate regressions. Specifically, probit or logit models were employed by more than half of the studies. As they are deeply rooted in economics, these econometric models were certainly not accidentally used in the examined rationalist empirical works. Both logistic regression and ordinary least squares (OLS) methods were the second most frequently used methods in the examined literature [63,77,80,86,87]. Third, a chi-square test and a cross-tabulation analysis were each performed once each [79,83]. Finally, Bernauer et al. [76] employed descriptive statistics and mainly relied on binary correlation analysis. The centralization of statistical correlation allows for a broad spectrum of enriching mathematical techniques. For example, Bayesian models of structural equations enable the identification of mediating or moderating effects between variables.

### 4.3. Object of Analysis

This third and last step is conducted in two different stages. First, how the authors address the framework provided by Koremenos, Lipson, and Snidal [2] is examined. Second, their outcomes and the ways in which they answer rational statements are shared.

### 4.3.1. The Authors' Approaches to the Rational Project

Initially, the literature was divided between the causes and consequences research lens. While 15 out of 21 dived into the nature of institutional design, 6 contributions [75–77,79,88,89] query about its implications. Secondly, a systematization was employed to identify the specific variables analyzed within the Koremenos, Lipson, and Snidal [2] framework. However, some articles addressed the rational project without specifying a variable or conjecture [62,82,89]. Other contributions aimed to address the complete logic of the rational design proposition [84,85]. These records constituted 23.81% of the sampled literature. The remaining records that included specific variables in their investigations represented 76.19% of the sample.

Although the authors within this second group could explore dependent or independent variables, the systematization revealed that all of them focused on dependent variables. In other words, when these authors performed quantitative research on the rational design of institutions, they were interested in specific design features. In fact, 6 out of the 16 authors in this group concentrated only on institutional characteristics. On the other hand, cooperation problems are never studied in isolation. In the remaining 10 studies, they were addressed along with dependent variables. The frequency with which the institutional design features and cooperation problems were addressed is depicted in Figures 3 and 4. Such frequency incorporates the causes vs. consequences distinction.

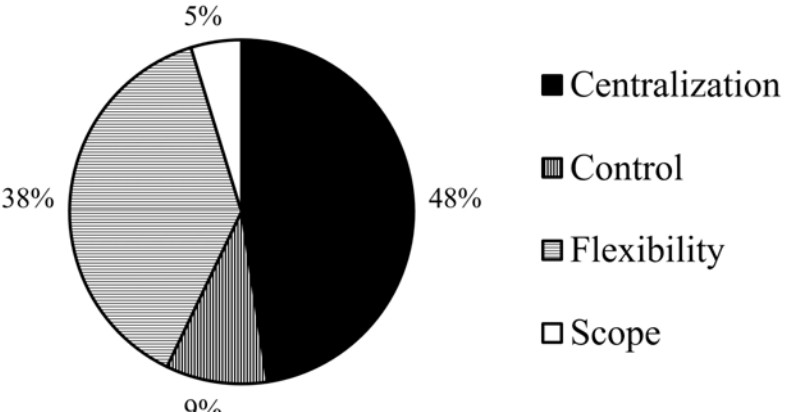

**Figure 3.** Frequency of institutional design features in the reviewed documents.

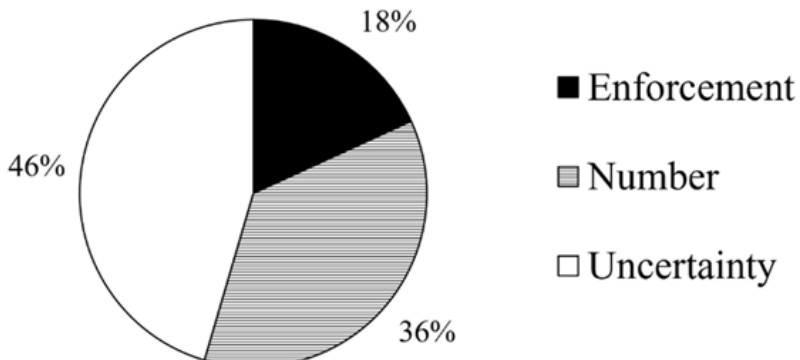

**Figure 4.** Frequency of cooperation problems in the reviewed documents.

In addition to these figures, Table 7 summarizes the way in which each author has dealt with the rational approach. The implications of the identified patterns are outlined in the following section. Within Table 7, a column has been placed to summarize the conjectures with which the authors deal, if any. Sometimes this inclusion was clearly expressed by the articles. In other occasions, articles hypotheses were related to rational design conjectures, but they did not mention them explicitly. Finally, some studies analysed variables without referring to rational project conjectures.

Regarding dependent variables, the systematization revealed a clear preference towards CENTRALIZATION and FLEXIBILITY. All of the corresponding conjectures were included in the quantitative research studies. In other words, the scholars who analysed the rational design of international institutions mainly focused on these two institutional features. The literature identified the concept of CENTRALIZATION in the context of the establishment of dispute settlement mechanisms and FLEXIBILITY in the context of the existence of escape and renegotiation clauses. Both of these design features are linked to the normal progress and development of an international institution once it has been established. Consequently, their prominence highlights how rationalism may hold higher explanatory power regarding how institutions develop and evolve.

The remaining dependent variables proposed by Koremenos, Lipson, and Snidal [2] received considerably less attention. The examined scholarship linked CONTROL with specific voting rules and SCOPE with the spectrum of issues addressed by institutions. This omission is consistent with the fact that both of these factors involve decisions traditionally addressed when institutions are founded. The same is the case with MEMBERSHIP, which is unrepresented in research studies and constitutes a gap in the literature.



**Table 7.** Summary of the authors' approaches.

| Code | Reference | Rational Logic | Dependent Variables | | | Scope | Independent Variables | | | Conjectures, If Any |
|------|-----------|----------------|--------------|---------|-------------|-------|-------------|--------|-------------|----------|
| | | | Centralization | Control | Flexibility | | Enforcement | Number | Uncertainty | |
| 001 | Allee and Elsig [74] | | 1 | | 1 | | | 1 | | F3 |
| 002 | Baccini, Dür, and Elsig [64] | | | | 1 | | | | | F3 |
| 003 | Bearce, Eldredge, and Jolliff [75] | | | | 1 | | | | | |
| 004 | Bernauer et al. [76] | | 1 | | 1 | | | 1 | | |
| 005 | Blake and Payton [77] | | | 1 | | | | 1 | | V1 |
| 006 | Copelovitch and Putnam [78] | | 1 | | 1 | | | | 1 | C2 and F1 |
| 007 | Hansen, McLaughlin Mitchell, and Nemeth [79] | | 1 | | | | | | | C3 and V1 |
| 008 | Hooghe and Marks [63] | | 1 | 1 | | | | 1 | | |
| 009 | Jetschke and Münch [80] | | 1 | | | 1 | | | | |
| 010 | Jo and Namgung [81] | | 1 | | | | | | | |
| 011 | Kaoutzanis, Poast, and Urpelainen [82] | 1 | | | | | | | | |
| 012 | Koremenos [36] | | | | 1 | | | | 1 | F1 |
| 013 | Koremenos [83] | | 1 | | | | 1 | | 1 | C1, C2 and C3 |
| 014 | Koremenos [84] | 1 | | | | | | | | V2 |
| 015 | Koremenos [85] | 1 | | | | | | | | C1, F1, F2, V2 and S3 |
| 016 | Kucik [86] | | | | 1 | | | | | |
| 017 | Lefler [87] | | 1 | | | | | | | |
| 018 | Marcoux [88] | | | | 1 | | | | 1 | |
| 019 | Mohrenberg, Koubi, and Bernauer [89] | 1 | | | | | | | | F1 and F3 |
| 020 | Tallberg, Sommerer, and Squatrito [62] | 1 | | | | | | | | |
| 021 | Tir and Stinnett [90] | | 1 | | | | 1 | | 1 | C2 and C4 |
| | Total | 5 | 10 | 2 | 8 | 1 | 2 | 4 | 5 | |

Regarding independent variables, UNCERTAINTY arose as the most frequently observed cooperation problem, followed by NUMBER. This observation is consistent with the predominant dependent variables and with rational logic. After all, institutions resolve adverse circumstances such as information asymmetries and uncertainties within inherently hostile environments to facilitate cooperation [85]. The other two examined difficulties received less attention. ENFORCEMENT only appeared in two records, and DISTRIBUTION was not mentioned in the quantitative studies. Considering that the literature has identified compliance with commitments as one of the recurrent problems of institutionalized cooperation [91], these trends reveal deficiencies in the rational design project and indicate future lines of research.

To facilitate a search for transversal patterns, Table 8 shows which variables were analyzed in each category of international institutions. The examined empirical studies prioritized FLEXIBILITY when analyzing agreements and CENTRALIZATION when studying organizations. The agreement research focused on procedures used to adapt to new circumstances, and the organization research focused on individual organizations' formalization levels or their performance of important tasks. This finding is consistent with the nature of these organizations as international institutions. The same pattern is applicable to the examined cooperation problems. The authors examined UNCERTAINTY when observing the design of agreements and NUMBER when considering an organization's design.

**Table 8.** Studied variables by type of institution.

| Approach | Institution | | Total |
|---|---|---|---|
| | **Agreement** | **Organization** | |
| Dependent Variables | | | |
| Centralization | 6 | 4 | 10 |
| Control | | 2 | 2 |
| Flexibility | 8 | | 8 |
| Scope | | 1 | 1 |
| Independent Variables | | | |
| Enforcement | 2 | | 2 |
| Number | 2 | | 4 |
| Uncertainty | 5 | | 5 |

### 4.3.2. Rationalism Contestation

Thus far, the relevance of the rational design of international institutions has been verified. The findings of this study on units of analysis and methodological approaches have shown the need for a complementary dialogue between traditions. Furthermore, the initial observation made regarding the studies' objects of analysis has revealed the existence of concrete rational research patterns supporting this suggestion. Accordingly, it should now be determined how these studies support rational statements. In doing so, this research aims to clarify the extent of the explanatory power of rationalism. The previous systematization categorized the examined articles according to their implications for rational design projects. These implications could support or complement Koremenos, Lipson, and Snidal's [2] approach.

As observed in Figure 5, the main group of the articles (10 out of 21) supports rationalism with empirical evidence. Remarkably, three of them are authored by Koremenos. Initially, this research suggests that Koremenos, Lipson, and Snidal's [2] statements are strongly supported by the empirical literature. The support of the second subgroup (11 out of 21) of studies is ambiguous, and the rational explanations of these studies are complemented by other potential variables. Noticeably, this trend supports the belief that institutionalism schools should meet halfway.

Aiming to offer depth in terms of the implications of these results, Table 9 categorizes the research outcomes according to the specific variables that each study examines. As shown, the quantitative research studies on the rational design of institutions strongly sup-

port conjectures related to certain variables such as CENTRALIZATION and FLEXIBILITY. However, even these are complemented with other explanatory variables. In the same line, research reveals how rationalism alone particularly struggles to explain how international institutions define their SCOPE and CONTROL features. Regarding cooperation problems, rationalism offers sound explanations for how decision-makers deal with UNCERTAINTY. However, other logics, such as constructivism or realism, complement rational schools of thought on how actors address ENFORCEMENT and NUMBER problems.

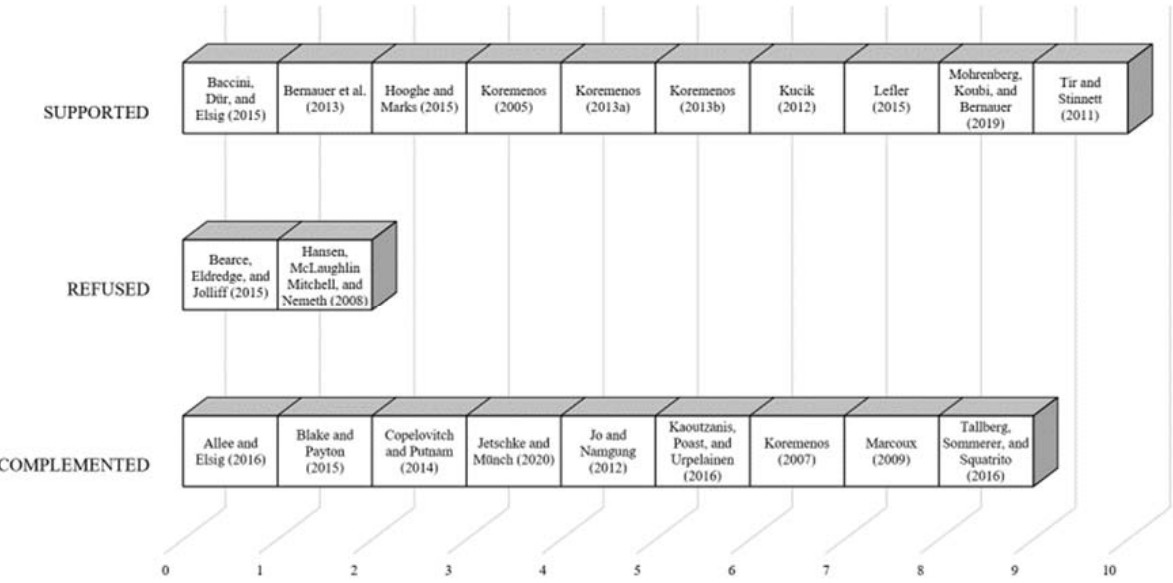

**Figure 5.** Articles that support, refuse or complement the rational approach.

**Table 9.** Research implications for rational variables.

| Approach | Outcome Complemented | Refused | Supported | Total |
|---|---|---|---|---|
| Dependent Variables | | | | |
| Centralization | 5 | 1 | 4 | 10 |
| Control | 1 | | 1 | 2 |
| Flexibility | 3 | 1 | 4 | 8 |
| Scope | 1 | | | 1 |
| Independent Variables | | | | |
| Enforcement | 1 | | 1 | 2 |
| Number | 2 | | 2 | 4 |
| Uncertainty | 3 | | 2 | 5 |
| Rational Logic | 2 | | 3 | |
| Total | 18 | 2 | 17 | 37 |

Regarding CENTRALIZATION, the rational conjectures are mostly complemented by other approaches such as diffusion literature. Thus, several articles argue against Koremenos, Lipson, and Snidal's [2] proposition even though their statements are verified. The earliest empirical research on CENTRALIZATION was developed by Koremenos [83]. This work discusses how cooperation problems such as UNCERTAINTY and ENFORCEMENT influence institutional CENTRALIZATION. Inspired by Goldstein et al. [51], Koremenos [84] includes COMMITMENT and the prisoners' dilemma-like incentives as cooperation as cooperation problems in this work. Remarkably, the inclusion of COMMITMENT represents an attempt to incorporate growing concerns within the International Law discipline into the rational project. As a result, it reconciles the International Relations and International Law bodies of literature on the concept of compliance, which is a dichotomy overviewed in previous works [91].

This new independent variable, which is conceptually close to ENFORCEMENT, is based on the understanding that rational actors use international legalization to constrain present and future domestic behavior. This use of international legalization is identified as increased centralization or the integration of delegated dispute settlement resolution procedures. Bélanger and Fontaine-Skronski's [52] (p. 258) empirical research on the concept of legalization supports this identification. Their work identifies three dimensions that constitute legalization: obligation, precision, and delegation. The latter dimension is related to the centralization feature of rational authors.

Consequently, Koremenos [83] represents a considerable advancement in the context of the CENTRALIZATION issue. Statistical patterns show that the decision to include dispute resolution within the governance structure of an international agreement is induced by a cost (delegation entitles sovereignty lost) and benefit (challenging cooperation problems) analysis. In other words, empirically, actors opt for high levels of centralization or legalization when they face UNCERTAINTY, ENFORCEMENT, and COMMITMENT problems. In line with this logic, Tir and Stinnett [90] determine that highly contentious issues have a greater effect on the institutional design of river treaties than contextual and power politics factors. They verify that implementation and compliance difficulties, such as UNCERTAINTY or ENFORCEMENT problems, are associated with high levels of INSTITUTIONALIZATION or the inclusion of provisions for institutional governance (centralization). In doing so, they support rational problem-solving logic and Bernauer et al.'s [76] statement that environmental problem structures are a key determinant of institutional design.

Tir and Stinnett [90], Jo and Namgung [81], Hansen, McLaughlin Mitchell, and Nemeth [79] and Lefler [87] conduct studies that deal with CENTRALIZATION. All of these researchers vaguely explore Koremenos, Lipson, and Snidal's [2] conjectures, but their conclusions inherently contribute to the examined debate. First, Jo and Namgung's [81] (p. 1061) research on preferential trade agreements reveals that both macro- and micro-level factors explain the inclusion of dispute settlement mechanisms. The micro-level incentives introduced by the rational design project are complemented by the macro-level trends within the literature on diffusion. Remarkably, their findings support legalization theses [3] by which the delegation of dispute settlement functions is subject to domestic politics as well as international political conditions. In their work, they note a crucial constructivist concern. They show that previous legal frameworks have an influence on subsequent institution creation, especially in the context of dispute settlement mechanism design in trade agreements. This emulation process occurs through comembership and is explained by the fact that decision makers are usually "trade bureaucrats" who participate in subsequent trade negotiations. This finding supports the belief that institutional designs are not only the result of problem-solving procedures; indeed, other explanatory variables, such as existing legal frameworks, also contribute to their creation.

Hansen, McLaughlin Mitchell, and Nemeth [79] (p. 300) also underline the role of executive and highly centralized bureaucratic branches in the promotion of dispute resolution. Their research addresses the role of international organizations as problem-solving structures. They connect their success as conflict managers to institutionalization (CENTRALIZATION), member preference similarities (a concept that reminds them of the distribution cooperation problem), and democratic member history. Their proposition applies rational design logic, according to which institutions are problem-solving structures. Such research, placed within the consequences approach, concludes with the need to include contingent factors such as preferences and domestic regimes when analyzing IO efficacy. Finally, Lefler's [87] (p. 93) analysis of international organizations focuses on diverse approaches to dispute resolution based on control, transparency, and elements of distributional bias. It examines why states strategically select mediation or arbitration for conflict management. It can be determined that this research supports rational logic, as it concludes that forum selection for dispute resolution is related to settlement compliance.

Instead of reviewing only CENTRALIZATION problems, Bernauer et al. [76], Copelovitch and Putnam [78], and Allee and Elsig [74] additionally analyze the issue of FLEXIBILITY. The first study supports a rational approach, while the other two complement it. The research of Bernauer et al. [76], based on the presumption that states are rational actors, proves the inconsistency of the depth versus participation dilemma in the context of global environmental cooperation. Copelovitch and Putnam's [78] research is similar since they take duration, dispute settlement clauses, and exit clauses into consideration. However, they discuss the explanatory power of deductive and rational design theories, claiming that strategic behavior should be examined in conjunction with the decision environments where such behavior occurs. As stated in this study, the institutional context jeopardizes the explanatory power of uncertainty because prior institutional and legal commitments also shape states' choices on agreement design. Notably, this statement is in line with the findings previously underlined. Similarly, Allee and Elsig's [74] (p. 116) research on international agreements concludes that their investigation "reconciles divergent perspectives on institutional design, showing that international agreements can reflect not only rational design but also the unique needs and preferences of states and regional actors".

On the subject of FLEXIBILITY, this research observes both supportive voices [2,64,86] and enlightening debates. For instance, Marcoux's [88] (p. 225) research on multilateral environmental agreements (MEAs) complements the rational approach. This study adds power asymmetry explanations to rational statements on flexibility provisions. Notably, this implies that, empirically, realist statements could also be incorporated into the design of international institutions. Bearce, Eldredge, and Jolliff [75] also complement the FLEXIBILITY proposition but with a consequential approach. By alternatively treating this feature as an independent variable, they show its impact on negotiation length.

CENTRALIZATION has also been studied in conjunction with CONTROL. Hooghe and Marks [63] (p. 307) connect these variables with the concepts of delegation and pooling, which are two dimensions of authority in international organizations. Thanks to authority, organizations can deal with problems associated with many members and large organizational scopes because it reduces transaction costs and centralizes decision-making procedures. These results support rational conjectures, showing that large membership groups (NUMBER) are positively related to CENTRALIZATION and CONTROL. Additionally, this research points out that Hooghe and Marks [63] (p. 311) partially argue in support of Koremenos, Lipson, and Snidal's [2] proposition since SCOPE, a dependent institutional feature, is identified as a driving factor of authority. Similarly, Jetschke and Münch [80] state that functionalist mechanisms are more consistent than domestic regimes in terms of conducting delegation procedures by which parliaments and courts are created. In other words, these mechanisms are initially created to monitor the compliance of member states, and eventually they may become institutions of democratic control. This research finds that the democratization literature adopts a particularly strong confrontational perspective on rational design, especially regarding its CONTROL features.

Additional quantitative research on CONTROL, such as that of Blake and Payton [77], found that the voting rules of states are explained by the design objectives that they prioritize: control, effective membership, compliance, or responsiveness. According to rational logic, when the NUMBER or membership is large, states avoid establishing unanimity voting rules (less CONTROL) to ensure responsiveness. On the other hand, they choose to implement unanimity rules (greater CONTROL) when their core interests are at stake and implement weighted voting to retain the major powers central to their institutions' effectiveness.

Once the ways in which authors address rational variables have been examined, the articles that focus on RATIONAL LOGIC as a whole should be observed closely. Koremenos [84,85] introduced the COIL database and researched the inclusion of punishment provisions that support rational design. The theoretical reflection of Koremenos [84,85] links cooperation problems such as enforcement, uncertainty, number, and commitment with the design of punishment provisions. These provisions may vary in terms of formality or flexibility depending on the heterogeneity and power asymmetry that involved

parties must address. This connection between flexibility and distribution problems involving power is also outlined by Marcoux [88]. In line with Marcoux [88], Mohrenberg, Koubi, and Bernauer [89] investigate the design of multilateral environmental agreements. This research offers empirical support to the problem-solving structures of international institutions that underline rationalism.

On the other hand, the research studies of Kaoutzanis, Poast, and Urpelainen [82] and Tallberg, Sommerer, and Squatrito [62] address rationalism in the context of the IO democratization literature. Kaoutzanis, Poast, and Urpelainen [82] examine rational logic in the context of the design of institutions but emphasize the role of domestic regimes in how states overcome cooperation problems. Tallberg, Sommerer, and Squatrito [62] go one step further. According to their research, dominant approaches such as rational functionalism and sociological institutionalism should be complemented by a third factor: regime type. In other words, cooperation problems (Koremenos, Lipson, and Snidal [2]), organizational culture (Barnett and Finnemore [22]), and domestic political systems together explain the design of international institutions.

## 5. Conclusions and Future Lines of Research

As global governance challenges increase, actors create formal and informal international institutions that they trust to address these issues, even if they do so imperfectly. Indeed, the literature on the design of international institutions is increasingly relevant for academics and decision-makers. This literature is dominated by opposed logics and a major debate between the constructivist and rational approaches. This research contributes to this debate by showing that, instead of being in opposition to one another, these perspectives are complementary.

This research focuses on a promising rational contribution: The Rational Design of International Institutions (Koremenos, Lipson, and Snidal [2]). This model contains a collection of conjectures that relate cooperation problems to the institutional design characteristics that they influence. Both extensive criticism and considerable support have been given in response to their proposition and participation in the institutional design debate. This research aims to observe whether and how rational design has been studied using quantitative research.

With a vector that integrates an object, unit, and methodological approach of interest, 161 articles are identified within the Web of Science database. Through a criteria-based process of screening and selection, a body of 21 studies is subjected to systematization. The fact that this research extracts articles from only one database, though it is extensive and well renowned, is its main limitation. Future research should retrieve records from alternative databases. Second, given that it is a systematic review, the findings of this study are mainly qualitative. In other words, a metanalysis on the design of international institutions would be even more enriching. Finally, limitations may abound in the data collection process. Relevant publications may be excluded from the search. In addition, future notable articles could be added to the rational design literature, rendering this work obsolete.

The findings of this study are articulated into three blocks. The findings regarding the studies' units of analysis and methodological approaches reveal gaps and insufficiencies in terms of the rational design of international institutions. This supports the need for a dialogue between traditions. First, the unit of analysis section focuses on the concrete international institutions that the examined authors analyze. Surprisingly, this research encounters a pre-eminence of informal institutions as agreements over formal institutions as organizations. The agreements differ in terms of issue area and organizational scope. Second, the methodological approach section focuses on the databases and research methods employed by the examined publications. The COIL and ICOW projects dominate the authors' datasets. Additionally, all the articles conduct cross-sectional studies and rely on statistical correlation to verify their hypotheses. Time variable observations are used to confirm some constructivist and historical perspectives regarding the rational inability to

consider contingent factors. Additionally, single-method testing is used rather than other valuable mathematical techniques for variable behavior observation.

The object of analysis section reveals how the authors approached Koremenos, Lipson, and Snidal's [2] proposition. The findings support the belief that the design of international institutions is equally explained by different schools of thought. Regarding how the publications address rational design projects, some only utilize rational logic testing, while others introduce concrete variable analyses. The concepts of CENTRALIZATION and FLEXIBILITY were the primary design characteristics analyzed, and UNCERTAINTY drew the most academic attention. These variables also received extensive support in the literature. This reveals how the explanatory power of rationalism concentrates on how states address UNCERTAINTY by introducing delegated dispute resolution mechanisms and escape or renegotiation clauses to international institutions. Rationalism has received extensive empirical criticism as well as empirical support. Some of its assumptions were verified but were supplemented with constructivist or domestic regime perspectives.

In short, the rational design research program offers the opportunity to make substantive advances on the problem of the design of institutions that facilitate international cooperation and guidance on what factors should be considered when designing institutions that aspire to achieve such objectives. However, it cannot guarantee their full effectiveness, as suggested by some of the main initial critics of this project (Wendt [27]). The classic problem of compliance involving the commitments of actors, generally states, in an international context of "anarchy" or the predominance of nation-state sovereignty remains an insurmountable obstacle. This is due to its structural, dynamic, and multi-causal nature, which is difficult to isolate because of the divergent, epistemologically differentiated logics involved.

Each of the two types of institutions, international agreements and international organizations, serves different functions. When the institution involves private, nongovernmental actors, international agreements are more operational and probably more efficient, but not more effective, as is the case with the 2030 Agenda. If the problem involves states, because of their scope, international organizations offer a more appropriate and legalized institutional environment. They have more legitimacy and relative authority. But the usual lack of coercive mechanisms limits their operational capacity, as has been the case with the World Health Organization and its inability to establish rational vaccination mechanisms against COVID-19.

Finally, this research considers that both initial objectives have been largely accomplished. Additionally, it offers a complex perspective on the study of the rational design of international institutions. The findings developed here contribute to the academic literature, as they identify lines for future research that could extend the radius of analysis to qualitative studies and even incorporate a comparative perspective of rational and constructivist projects.

**Supplementary Materials:** The following supporting information can be downloaded at: https://www.mdpi.com/article/10.3390/su14137866/s1. File S1: PRISMA 2020 Checklist.

**Author Contributions:** Conceptualization, A.T.-V., F.S.-C. and A.S.; methodology, L.A.F.-P. and A.S.; formal analysis, A.T.-V. and F.S.-C.; investigation, A.T.-V. and A.S.; resources, F.S.-C.; data curation, F.S.-C.; writing—original draft preparation, A.T.-V. and L.A.F.-P.; writing—review and editing, A.T.-V., F.S.-C. and A.S. All authors have read and agreed to the published version of the manuscript.

**Funding:** This research received no external funding.

**Conflicts of Interest:** The authors declare no conflict of interest.

## Appendix A

This research followed the PRISMA (Preferred Reporting Items for Systematic Reviews and Meta-Analyses) approach, whose flow diagram (Moher et al., 2009) is presented in Figure A1.

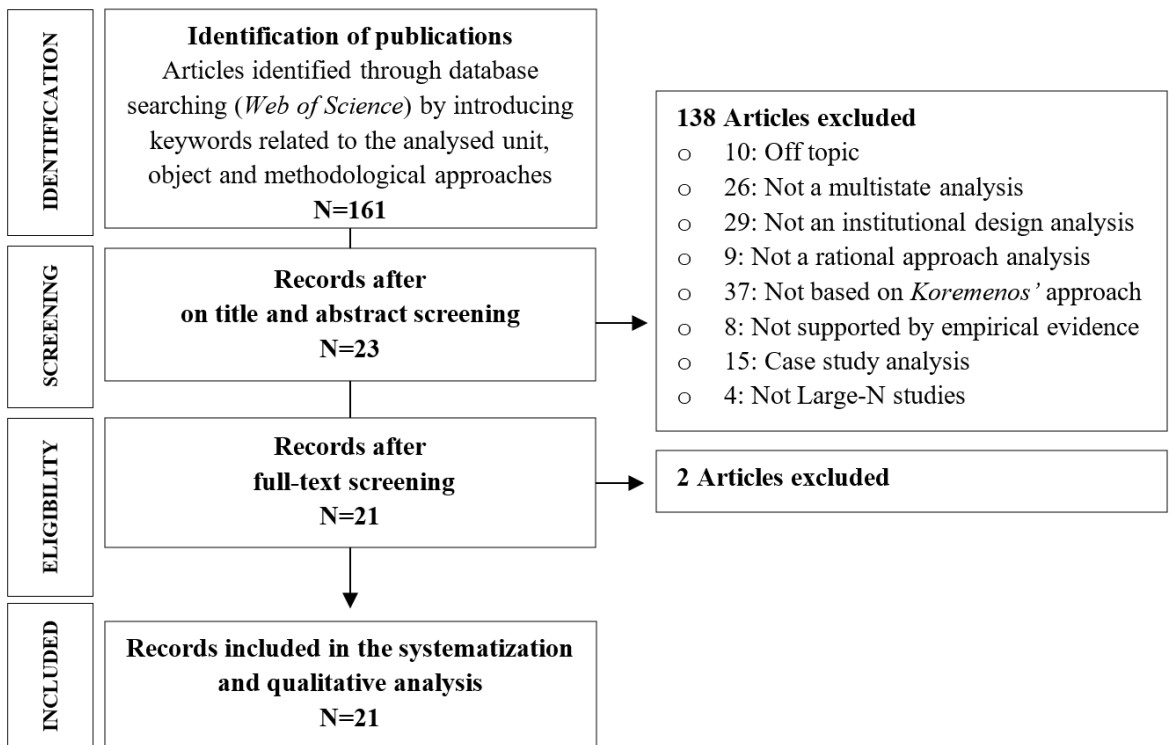

**Figure A1.** The PRISMA model.

The search vector used was TS = ("KOREMENOS" OR (("INSTITUTIONAL DE-SIGN" OR "DESIGN OF INTERNATIONAL INSTITUTIONS" OR "INSTITUTIONAL CHOICE" OR "RATIONAL CHOICE" OR "RATIONAL DESIGN" OR "RATIONALLY DESIGNED" OR "DESIGN CHARACTERISTICS" OR "DESIGN FEATURES") AND ("INTERNATIONAL INSTITUTIONS" OR "INTERNATIONAL ORGANIZATIONS" OR "INTERGOVERNMENTAL ORGANIZATIONS" OR "INTERNATIONAL AGREEMENTS" OR "INTERNATIONAL TREATIES" OR "GOVERNANCE INITIATIVES" OR "INTERNATIONAL RELATIONS") AND ("DATABASE" OR "DATA" OR "DATASET" OR "LARGE-N" OR "CASE" OR EMPIRICAL* OR MODEL* OR QUANTITATIV*)).

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
