# Peer review of "Can the Rational Design of International Institutions Solve Cooperation Problems? Insights from a Systematic Literature Review"

_sustainability, doi:10.3390/su14137866_

Round 1

Reviewer 1 Report

Sorry, but I find the article not only difficult to follow, but also unclear in its objective and content. It is just a (very partial) survey of some papers, but the claim is that it would be an empirical paper, only because it groups them in some ways, draws some graphs and calculates some percentages. I am sorry to have to report such a tranchant opinion, but I don't think that a scholarly journal should publish anything like this. 

Author Response

Dear reviewer, please see the attachment with our response to your comments. Thank you very much.

Reviewer 2 Report

I enjoyed this article on nature of institutions, their types and the outstanding methodology employed in relation to literature review. Never the less, there needs to be a clearer understanding of why? From a practical perspective, what value does this research add? Why is this research needed?  Perhaps provide examples of real life institutional  inefficiencies and how your research can address these inefficiencies

Author Response

(The authors gave the same response as above.)

Reviewer 3 Report

The main comment, which I have to share with the authors of "Can the rational design of international institutions solve cooperation problems? Insights from a systematic literature review", refers to the object itself of their study.

More precisely, despite understanding that their work concerns a systematic literature review, I feel that the reader does not get sufficient information on the international institutions (sometimes, called by them organisations) actually involved in their study. When have they been founded and where are they located? With which scopes and involving how/which member countries? These are crucial questions to be answered before moving to the analysis itself.    

Author Response

(The authors gave the same response as above.)

Reviewer 4 Report

Dear author(s) the topic is very original i really appreciate you study and your approach in order to identify the effect of international institution in relation to cooperations problems.

But i have some major concern about the structure of this paper:

In the introduction section I suggest to introduce better the international institutions and following explain the definitons of Koremenos, Lipson, and Snidal.

In the methodology i suggest to rehorganized the structure of these section, in my opinion, first of all, is more important to explain the definition of the sample with the PRISMA Model and the methodology of qualitative analysis.

Author Response

(The authors gave the same response as above.)

Round 2

Reviewer 4 Report

ell dove! In my opinion the paper is suitable for a pubblication.